# Sublethal Effects of Arsenic on Oxygen Consumption, Hematological and Gill Histopathological Indices in *Chanos chanos*

**DOI:** 10.3390/ijerph182412967

**Published:** 2021-12-08

**Authors:** Kannayiram Muthukumaravel, Kumara Perumal Pradhoshini, Natarajan Vasanthi, Venkatachalam Kanagavalli, Mohamed Ahadu Shareef, Mohamed Saiyad Musthafa, Rajakrishnan Rajagopal, Ahmed Alfarhan, Anand Thirupathi, Balasubramani Ravindran

**Affiliations:** 1Faculty of Sports Science, Ningbo University, Ningbo 315211, China; kumar_phd_2003@yahoo.co.in; 2P.G. and Research Department of Zoology, Khadir Mohideen College, Bharathidasan University, Adirampattinam 614701, Tamil Nadu, India; vasanthidr4@gmail.com; 3Unit of Research in Radiation Biology & Environmental Radioactivity (URRBER), P.G & Research Department of Zoology, The New College (Autonomous), University of Madras, Chennai 600014, Tamil Nadu, India; spradho@gmail.com; 4Department of Zoology, A.V.V.M Sri Pushpum College, Poondi, Thanjavur 613503, Tamil Nadu, India; kanagavalliapp83@gmail.com; 5P.G. and Research Department of Chemistry, The New College (Autonomous), University of Madras, Chennai 600014, Tamil Nadu, India; jasshaali@gmail.com; 6Department of Botany and Microbiology, College of Science, King Saud University, P.O. Box 2455, Riyadh 11451, Saudi Arabia; rrajagopal@ksu.edu.sa (R.R.); alfarhan@ksu.edu.sa (A.A.); 7Department of Environmental Energy and Engineering, Kyonggi University, Youngtong-gu, Suwon 16227, Gyeonggi-do, Korea

**Keywords:** *Chanos chanos*, arsenic, LC_50_, oxygen consumption, hematology, gill histology

## Abstract

Background: The current study was performed aiming to evaluate possible changes in the effect on oxygen consumption, hematology and gill histopathological parameters in fish (*Chanos chanos*) upon exposure to sublethal concentration of the metalloid arsenic. Methods: Bioassay tests were conducted for determining the LC_50_ values of arsenic for 96 h. Oxygen consumption in control and arsenic-exposed fish was estimated using Winkler’s method. Red blood corpuscular (RBC) count was examined with a Neubauer counting chamber under a phase contrast microscope. Hemoglobin (Hb) was estimated following the acid hematin method. Histopathological studies were carried by processing and staining the gill tissues with hematoxylin and eosin in accordance with standard histological techniques. They were then subjected to examination under a scanning electron microscope. Results: *Chanos chanos* exposed to 1/10th of LC_50_ (24.61%) for a period of 30 days exhibited a maximum decline in the rate of respiration, followed by a decline in RBC and Hb above 45.59% and 51.60%, respectively. Significant toxic lesions encompassing fused gill lamellae, detached gill epithelium, hyperplasia and hypertrophy of respiratory epithelium became heavy handed on the 30th day. Conclusion: Information synthesized from our study serves to be useful in monitoring and managing (As) contamination in the aquatic environment.

## 1. Introduction

The hydrosphere, wrapping about 71% of planet Earth, is under constant threat as a result of the manifold activities of mankind [1]. Exorbitant industrial expansion, in addition to technological advancements are major concerns for water pollution, making it one of the burdens of the modern era. Although, awareness pertaining to toxic substances being deliberately dumped into the environment is growing day by day, the industrial effluents and domestic wastes discarded into the aquatic environment still represent chief sources of pollutants toxic to fishes [1]. It is these contaminants of water bodies that cause detrimental effects to the aquatic fauna because of their physical and chemical nature [2,3,4]. They bring about rapid, grievous effects by changing physio-chemical parameters, followed by chronic effects as a result of the gradual accumulation of materials causing changes to ecological conditions as a whole, besides also causing sublethal effects and mortality in aquatic fauna such as prawns, crabs and fishes. The incorporation of relatively insignificant amounts of toxic materials also have a negligible effect on both community metabolism and water quality. Changes in the physiological and behavioral activity of organisms observed serve as extremely sensitive indicators for the sublethal effects of pollutants. Immense physiological, biochemical and histochemical changes have been reported in fishes exposed to a variety of pollutants [5].

Among the pollutants, heavy metals that have been introduced into the hydrosphere so far play a key role in the changes studied [6]. Arsenic metalloid, popularly called as the “King of poisons”, accumulates in large amounts in the environment due to leaching from pesticide, combustion of arsenic-containing fossil fuels, smelter run off and mine tailings besides extensive industrial activities and anthropogenic causes through sewage disposal [7]. Even though traces of certain heavy metals are mandatory for normal metabolic processes in animals, the existence of these metals in an elevated concentration proves to be lethal or sublethal to aquatic organisms [8]. This toxicity is not an intrinsic property of metals but is a manifestation of their interaction with living systems. The ability of arsenic to significantly affect growth performance and hematological parameters, leading to toxicity, was observed in *Platichthys stellatus* for the exposed dose of 600 µg/L [7]. The potential of arsenic to get absorbed through gills and produce perturbation in its subsequent antioxidant system was reported in a promising vertebrate model for *Danio rerio* upon exposure to a high dose of 100 µg/L [9]. Concerning the *Chanos chanos* fish, it holds a remarkable place in the aquaculture domain for its growing demand as it serves as one of the most important sea foods in many Southeast Asian countries for its mild sweet flesh. Accordingly, several toxicological studies were carried out in *Chanos chanos* by using endosulfan [10], *Alexandrium minutum* [11], potassium permanganate [12], zinc [13] and unionized ammonia [14]. Recent research by [15] evaluated the histological changes in gills and variations in osmoregulation upon exposure to heavy metals Cd and Cu in *Chanos chanos*. The current investigation, therefore, keeping in mind the economic potential of this fish, aims to evaluate histological changes in gills and consequent variations in oxygen consumption upon exposure to the metalloid arsenic.

The potential of fishes to serve as a staple source of protein in a nation’s diet calls for the need to design and establish novel approaches for the benefit of fishery development upon gaining a better understanding of this topic. It is noteworthy that the toxicological studies of fishes have focused the attention of researchers, seeing the remarkable emphasis and dependence on aquaculture in addition to the existing elevated awareness on hazardous water pollution [15]. Notable investigations carried out in fishes so far have involved various physiological, biochemical and histological parameters upon exposure to heavy metals. A study on the bioaccumulation and tissue distribution of arsenic in *Oreochromis mossambicus* intending to ascertain the human health risk assessment revealed the deposition of the heavy metal in target organs following the sequence intestine > liver > gill > muscle [16]. Although, arsenic and other heavy metal absorption commences in the key survival organ gills essential to gaseous exchange, the liver and kidney grab maximum attention owing to their detoxification, nitrogenous waste excretion and homeostasis function. However, a study established in *Clarias batrachus* reported 90% of arsenic accumulation in gills in comparison to other organs [17]. Hence the current study has been designed (i) to assess the changes in oxygen consumption following the standard procedure of Winkler’s method; (ii) to evaluate the alterations in hematological parameters–red blood corpuscular count using hemocytometer and hemoglobin (Hb) values using acid hematin method and (iii) to evaluate the histopathological changes harbored in gills by employing light and scanning electron microscopic techniques, upon exposure to arsenic in the most preferred edible estuarine fish, *Chanos chanos*.

## 2. Materials and Methods

### 2.1. Determination of LC_50_ Values

Milkfish, *Chanos chanos*, (weight 16 ± 1 g and length 11 ± 1 cm) were collected from Agniar estuary (Lat. 10°20′ N; Long. 79°23′ E) near Adirampattinam, southeast coast of India. The fishes were stocked in large cement tanks, washed and sterilized with potassium permanganate solution. Experimental fishes were maintained in small plastic troughs (20 L capacity). Physico-chemical parameters of tank water during trial conditions included: temperature: 24.5 ± 0.5 °C; pH: 8.3 ± 0.26; salinity: 17.0 ± 1.15 ppt; dissolved oxygen: 5.24 ± 0.1 mg/L. The water was changed frequently from the tanks set up in the laboratory in order to ease the burden of fish being sensitive to ammonia build up. Sodium Arsenite (Na_2_AsO_2_) purchased from (Loba Chemie Pvt Ltd., Mumbai, India) was used in the present study. Bioassay tests were carried out for the determination of LC_50_ values for 96 h in accordance with the methodology of [4] where, different concentrations of arsenic (1, 2, 3, 4, 5, 6, 7, 8, 9 and 10 ppm) were prepared from the stock solution (1 ppt) enlisted in the Table 1. A set of ten fishes in triplicates, were introduced into the segregated experimental concentrations. Physico-chemical parameters of tank water during experimental conditions included: temperature: 24.5 + 0.5 °C; pH: 8.4 ± 0.21; salinity: 17.0 ± 1.15 ppt; dissolved oxygen: 5.16 ± 0.1 mg/L. Following exposure, fishes were randomly selected after 30 days where studies on oxygen consumption, hematological and histological changes were carried out with an interval period of 10 days.

### 2.2. Estimation of Oxygen Consumption

A series of rectangular glass jars containing water with a holding capacity of one liter maximum was set up for serving as aquaria for the fish. Care was taken to prevent the trapping of air bubbles. Each aquarium was administered by an individual and a thick layer of coconut oil was spread on the surface of the medium to curtail its contact with atmospheric air, as suggested in the standard procedure of Winkler’s method. Prior to the commencement of experiments, the initial oxygen content of water utilized for filling the aquaria was estimated following Winkler’s method [18]. A healthy fish was then allowed to respire for one hour in the destined animal chamber (aquarium). After one hour, water samples from the subsequent respiratory chamber were siphoned out into a bottle of known volume for evaluating the dissolved oxygen using the following formula:$$(O_{2})\;(\mathrm{mg}/L)=\frac{\left(V1\times N\times E\times 1000\right)}{\left[\frac{V4\left(V2-V3\right)}{V2}\right]}$$
where,

*V*1 = Volume of the titrant sodium thiosulphate (Na_2_S_2_O_3_) (mL),

*V*2 = Total volume of the water sample (mL),

*V*3 = Volume of (magnesium sulphate (MnSO_4_) + potassium iodide (KI)) added (mL),

*V*4 = Volume of analyte taken in titration,

*E* = Equivalent weight of O_2_,

*N* = Normality of Na_2_S_2_O_3_ solution,

### 2.3. Preparation of Blood Smear

Blood samples were procured from live fish by puncturing the caudal vein with aid of a disposable hypodermic syringe accompanying a 20-gauge needle followed by instantaneous preparation of blood smear. Smears made from blood samples were air dried for 1 h and then fixed in 95% methanol at 4 °C followed by staining protocol comprising the use of Leishman’s stain and glycerol. The blood corpuscles were examined and photographed by phase contrast microscopy.

### 2.4. Estimation of Hemoglobin and Red Blood Corpuscular (RBC) Count

Standard procedures were followed for determining the red blood corpuscular (RBC) count and hemoglobin (Hb) values. Red blood corpuscular (RBC) count was accomplished with a Neubauer counting chamber, following the protocol of [19]. Hemoglobin content of the control and arsenic-exposed fish were estimated following the acid hematin method [20].

### 2.5. Histology

For histological studies, tissues harvested from the gill region of 10 fish samples were subjected to a sequence of processes such as fixation, dehydration, embedding, sectioning followed by staining with hematoxylin–eosin dye by adopting the procedure of [21]. The stained slides were then covered with a coverslip to examine them under the microscope. Observed histopathological lesions in the specimen were evaluated under different magnifications and were photographed.

### 2.6. Scanning Electron Microscope (SEM) Studies of Gill

For the characterization of the affected gill tissue from arsenic-treated fish, SEM studies were performed where samples of tissue collected were washed repeatedly in 0.2 M phosphate buffer solution and then fixed in 3% glutaraldehyde. Then the tissue was dehydrated in acetone grades followed by critical point drying. Dried gills were mounted on the stub and were sputter-coated with gold in a gold coating unit (Thickness 100A). The tissue was then examined and photographed by using JEOL JSM 6360 scanning electron microscope (SEM, Tokyo, Japan) [22].

### 2.7. Statistical Analysis

The data obtained from the study, including O_2_ consumption, hematological parameters such as RBC and hemoglobin, were subjected to statistical analysis using SPSS software computer version 2015 (IBM, Armonk, New York, NY, USA) and one-way ANOVA. Values are reported as mean, standard deviation and standard error. Significant differences are presented in (*p* < 0.05) level. Acute toxicity (96 h LC_50_) tests were statistically analyzed using Finney’s method of Probit static bioassay test.

## 3. Results

### 3.1. LC_50_ Values of Arsenic

In the present study the LC_50_ values of arsenic for *Chanos chanos* at 24, 48, 72 and 96 h were found to be the same by arithmetical and graphical interpolation, i.e., 3.649, 3.185, 2.890 and 2.24 ppm, respectively, listed in Table 1.

### 3.2. Rate of Oxygen Consumption

The results of oxygen consumption of *Chanos chanos* fish exposed to 10% sublethal concentration of arsenic against the controls are presented in Figure 1 where a decline in the O_2_ consumption rate was observed following an exposure period of 10, 20 and 30 days. The rates of oxygen consumption in control *Chanos chanos* were 0.632, 0.641 and 0.638 mL O_2_/g/h at 10, 20 and 30 days, respectively. The fish exposed to sublethal concentrations, on the other hand, were recorded with relatively low rates of oxygen consumption including 0.580, 0.542 and 0.481 mL O_2_/g/h at 10, 20 and 30 days, respectively. From the study performed, the oxygen consumption rate was inferred to decrease gradually with increasing exposure periods. A maximum decline of about 24.61% in the rate of respiration in comparison to controls was noticed 30 days after exposure.

### 3.3. Changes in the RBC and Hb Content of Chanos Chanos to Sublethal Concentration of Arsenic

A prominent decline in the total RBC count of *Chanos chanos* were visualized for the dose of 10% sublethal concentration as presented in Figure 2. A marked decrease in the number of red blood cells of about 27.69, 36.97 and 45.59% was recorded at 10, 20 and 30 days of exposure. The hemoglobin content for the same dose concentration of arsenic showcased a decreasing trend with a marked reduction of about 20.69, 35.75 and 51.60% at 10, 20 and 30 days, respectively, as presented in Figure 3. Moreover, the current study recorded certain anomalies in the red blood cells of arsenic-exposed fish, which included abnormal cell shape, vacuolation, swelling and disintegration, as presented in Figure 4.

### 3.4. Histological Study

#### 3.4.1. Control Gill, Light and SEM Microscopic Observation

In *Chanos chanos*, the secondary gill lamellae of control fish appeared as finger-like projections with countless vascular cells being covered with a thin epithelial layer. They were very thin and slender, attaching on either side of the primary gill lamellae (Figure 5a–d).

#### 3.4.2. Histological Changes of Gill Tissue Induced by Arsenic under Light Microscopic Observation

Multiple histopathological lesions appeared in the gills of arsenic-treated fish. The secondary lamellae became degenerated and disintegrated. Most of the secondary lamellar layers were found to be completely ruptured, with hypertrophy. Hyperplasia with lamellar fusion in addition to vacuolation, were also discerned in the secondary lamellae of all treated fishes (Figure 5a–c).

#### 3.4.3. Histological Alterations of Gill in Arsenic-Treated Fish under SEM Observation

Upon exposure to the 10% sublethal concentration of arsenic for a period of 10 and 20 days, a fusion of secondary lamellae was observed (Figure 6b,c). The damage perceived was critical and progressive in post 30 days treated fish. The changes detected included deformations in primary and secondary lamellae with fusion between adjacent lamellae followed by curling and degeneration of secondary lamellae (Figure 6d).

## 4. Discussion

The LC_50_ value for the procured specimen in the current study was recorded to be 2.25 ppm, remaining constant upon exposure to increasing hours. Fish gills contribute to the key function of respiration and osmoregulation. Proximal contact of this organ with the external environment harboring various contaminants [23,24] make it a target organ for the reported uptake of toxicants and subsequent pathological manifestations. Since the rate of oxygen consumption has been considered as a true index of the overall metabolic state of animals, the change in respiratory activity can be used as an indicator of stress in animals exposed to toxicants [25]. Accordingly, the current study reports relatively low uptake of oxygen at a rate of 0.481 mL/g/h in the affected fish upon exposure to sublethal concentrations of arsenic against the control (0.638 mL/g/h) in a month’s period. This can be correlated to gill damage or to the inhibition of the mitochondrial enzyme system indicating the onset of severe hypoxia under heavy metal stress which had subsequently triggered the metabolic pathways in the current study. Similar results showcasing decreased oxygen uptake were reported in *Cyprinus carpio* [26], *Oncorhynus mykiss* [27] and *Cyprinus carpio* larvae upon exposure to copper [28]. Effects drawn out upon exposure to heavy metals comprising mercury, copper and cadmium, in *Labeo rohita* [29] and in *Cyprinus carpio* due to copper, cadmium and hexavalent chromium, harbored significant changes pertaining to a reduction in the RBC count [30]. In addition, the exposure of iridescent shark, *Pangusius hypophthalmus* [31] and *Channa striatus* to the heavy metal lead and metasystox, respectively, accounted for the injury caused to RBC cells. Thus, the results of the study are in accordance with the earlier report by [32] on oxygen consumption by fish in heavy metal contaminated water.

From the study undertaken it was also remarkable to note the conferred changes in *Chanos chanos*, which developed irregular erythrocytes with vacuolization, fragility, wrinkled membrane and hemolysis, as presented in “Figure 4”. Besides this, the total red blood corpuscle (RBC) count and hemoglobin (Hb) content reduced significantly to 45.59% and 51%, respectively, at the end of 30 days in the exposed fish with (*p* < 0.05). Results of the study are in congruence with the results obtained from studies on *Clarias*
*gariepinus* treated with lead [33] and with the investigations by [34,35,36,37] upon exposure to heavy metals. Moreover, they also support the previous findings in *Clarias batrachus* and *Lates calcarifer*, displaying similar decrease in values of blood parameters upon exposure to the heavy metals cadmium and mercury [34]. Thus, the lower RBC count and hemoglobin (Hb) levels of arsenic-treated *Chanos chanos* in the present study can be attributed to the disruption of iron synthesizing machinery as suggested by [38].

The histopathological changes observed in the gill tissue encompass fused secondary lamellae, degeneration of gill epithelium, hypertrophy of gill filament, hyperplasia of epithelial surface and necrosis as presented in Figure 5 and Figure 6. Results of the current study were found to correspond with similar findings in *Catla catla* [39], *Labeo rohita* exposed to aluminum chloride [40] and in gills of *Puntius ticto* [41]. Moreover, the results displayed are in accordance with findings by [42] who observed the direct biological effects of trivalent forms of arsenic on *Channa punctatus* upon exposure to a sublethal dose of 12.8 mg/L.

To conclude, the current study, which significantly focused on oxygen consumption, hematological and gill histopathological parameters, proves arsenic to be moderately toxic to *Chanos chanos,* following the entry of metalloid arsenic into the hydrosphere from both natural and anthropogenic sources. Similar sublethal effects brought about by this heavy metal in the aquatic environment call for the need to implement well organized mitigation measures. Their goal would be to prevent the currently alarming detrimental consequences imposed on aquatic health, particularly in fish. Predators, including humankind, are likewise in danger of getting exposed upon consumption of these arsenic-affected fishes. The current study has been implemented to evaluate the effects of arsenic to further encourage future research to be carried out regarding human health risks and in the hopes that novel mitigation measures will be designed to reduce heavy metal toxicity, especially arsenic which prevails in aquatic environments.

## 5. Conclusions

The present study reports significant alterations in hematological parameters (RBC & Hb) (*p* < 0.05) and histopathological lesions, such as fused secondary lamellae, degeneration of gill epithelium, hypertrophy of gill filament, hyperplasia of epithelial surface and necrosis in the gills of arsenic-exposed *Chanos chanos*. This indicates reduced oxygen consumption (*p* < 0.05) by them as a consequence, against the controls. Such drastic differences in the aforementioned indices between the control and experimental fish indicate that fishes are extremely sensitive to arsenic toxicity and that their mortality rate is both dose- and time-dependent. Relating to contamination in the natural environment, consumption of arsenic-affected *Chanos chanos* will inevitably expose humans to the risk of developing deleterious effects as concentration exceeding the natural limit is toxic. Therefore, information synthesized from this current study serves to be useful for monitoring and managing contamination in the aquatic environment.

## Figures and Tables

**Figure 1 ijerph-18-12967-f001:**
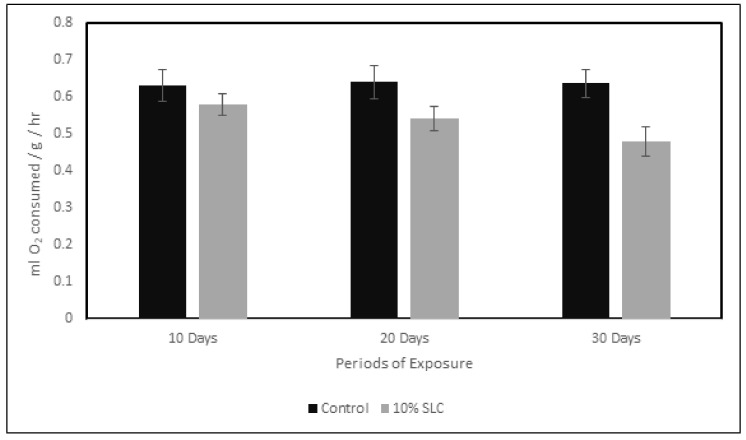
Oxygen consumption (O_2_ consumed mL/g/h) of *Chanos chanos* upon exposure to the 10% sublethal concentration of arsenic. Errors of the vertical bars denote standard error. Significant difference (*p* < 0.05) was observed between the controls and in fish at 20th and 30th day post arsenic exposure. (*p* > 0.05) between the control and fish at 10th day post arsenic exposure.

**Figure 2 ijerph-18-12967-f002:**
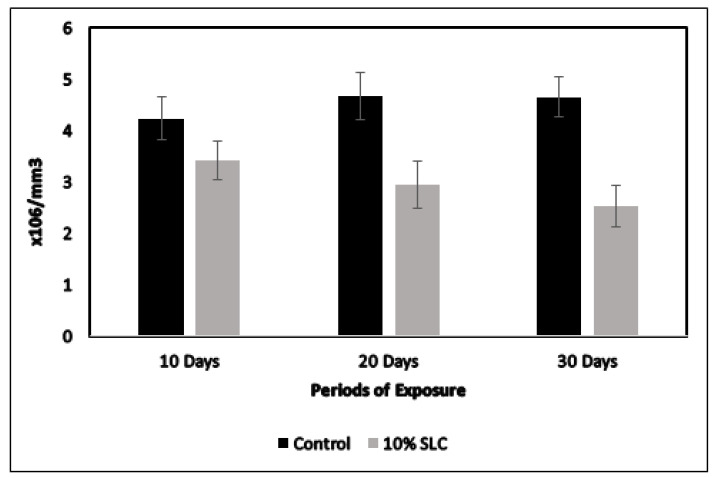
Total RBC count of *Chanos chanos* upon exposure to the 10% sublethal concentration of arsenic. Errors of the vertical bars denote standard error. Significant difference (*p* < 0.05) was observed between the controls and in fish at 20th and 30th day post arsenic exposure. (*p* > 0.05) between the control and fish at 10th day post arsenic exposure.

**Figure 3 ijerph-18-12967-f003:**
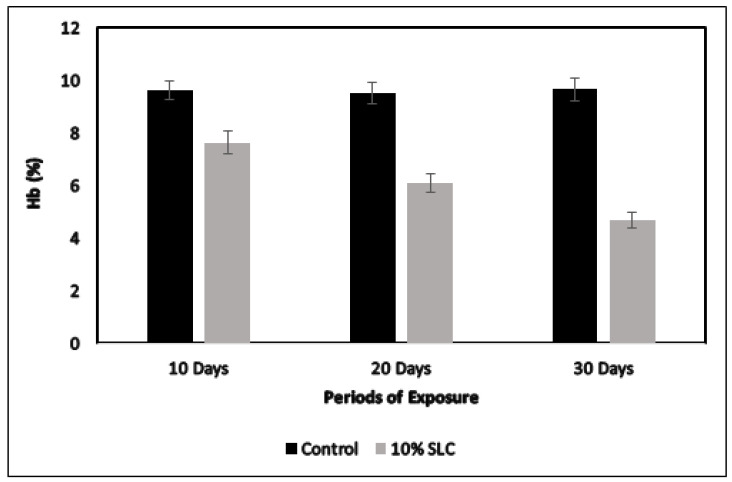
Hemoglobin content of *Chanos chanos* upon exposure to the 10% sublethal concentration of arsenic. Errors of the vertical bars denote standard error. Significant difference (*p* < 0.05) was observed between the controls and in fish at 20th and 30th day post arsenic exposure. (*p* > 0.05) between the control and fish at 10th day post arsenic exposure.

**Figure 4 ijerph-18-12967-f004:**
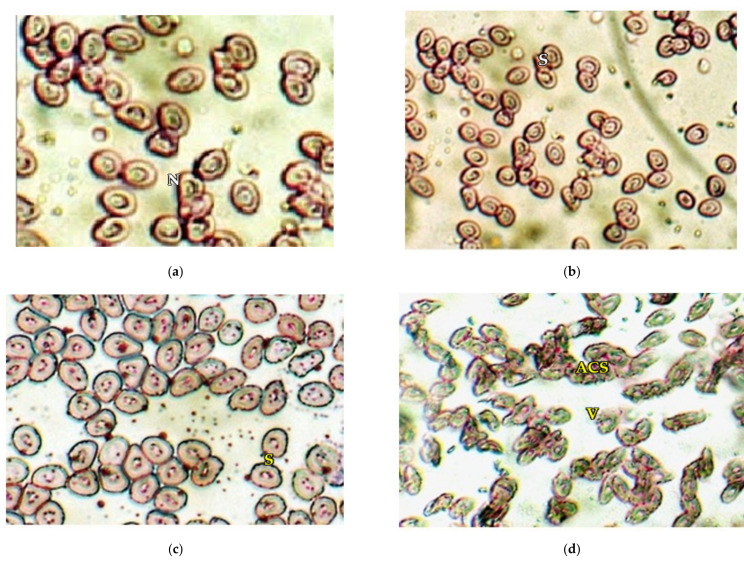
(**a**) Morphology of RBC in control fish (X 240) N—nucleus. (**b**) Morphology of RBC in arsenic-exposed fish after 10 days (X 100) S—swelling. (**c**) Morphology of RBC in arsenic-exposed fish after 20 days (X 450) S—swelling. (**d**) Morphology of RBC in arsenic-exposed fish after 30 days (X 450) ACS—abnormal cell shape, V—vacuolation.

**Figure 5 ijerph-18-12967-f005:**
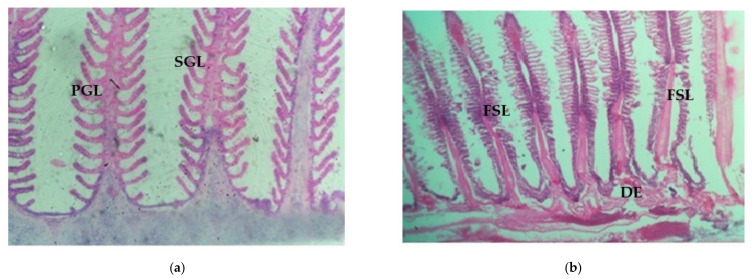
(**a**) Gill of control fish, PGL—primary gill lamellae; SGL—secondary gill lamellae; (**b**) Gill of 10% SLC of arsenic-treated fish (10 days); FSL—fusion of secondary lamellae; ESL—erosion of secondary lamellae; DE—degeneration of epithelium; (**c**) Gill of 10% SLC of arsenic-treated fish (20 days); FSL—fusion of secondary lamellae; H—hypertrophy; V—vacuolation; N—necrosis; DE—degeneration of epithelium; (**d**) Gill of 10% SLC of arsenic-treated fish (30 days); FSL—fusion of secondary lamellae; H—hypertrophy; N—necrosis; DLE—disintegration of lamellar epithelium; DSL—degeneration of secondary lamellae.

**Figure 6 ijerph-18-12967-f006:**
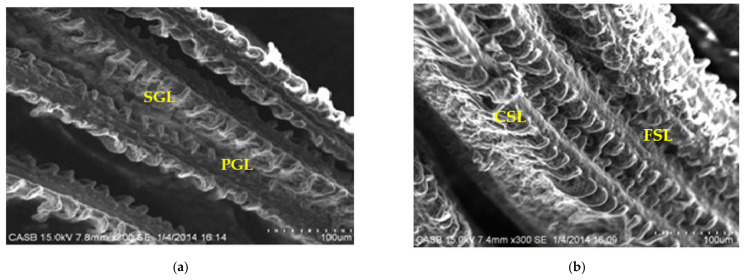
(**a**) SEM view of gill in control fish (PGL—primary gill lamellae; SGL—secondary gill lamellae); (**b**) SEM view of gill in 10% SLC arsenic-treated fish (10 days) (FSL—fusion of secondary lamellae; CSL—curling of secondary lamellae); (**c**) SEM view of gill in 10% SLC arsenic-treated fish (20 days) (FSL—fusion of secondary lamellae; DSL—degeneration of secondary lamellae); (**d**) SEM view of gill in 10% SLC arsenic-treated fish (30 days) (FSL—fusion of secondary lamellae; CPL—curling of primary lamellae; DSL—degeneration of secondary lamellae).

**Table 1 ijerph-18-12967-t001:** Percent mortality of *Chanos chanos* exposed to different concentrations of arsenic for different periods (ppm).

Hours of Exposure	LC_50_	L.C.L	U.C.L	Regression Equation	Calculated X^2^ Value	Table X^2^ Value
24	3.649099	4.0547773	0.284008	Y = 1.491007 + 6.241699X	13.9002	9.49
48	3.185816	3.661036	2.772282	Y = 2.731714 + 4.507538X	19.97842	11.07
72	2.890422	3.139665	2.660965	Y = 3.6252 + 2.982463X	1.457794	11.07
96	2.247149	2.464002	2.04938	Y = 4.172045 + 2.354607X	8.465149	12.59

Abbreviations and notations: LC_50_—lethal concentration required to kill 50% of the population; UCL—upper confidence limit; LCL—lower confidence limit (present study recorded 95% confidence levels, which is ideal).

## Data Availability

Not applicable.

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
