# Peer review of "Sublethal Effects of Arsenic on Oxygen Consumption, Hematological and Gill Histopathological Indices in Chanos chanos"

_ijerph, 2021, doi:10.3390/ijerph182412967_

Round 1
Reviewer 1 Report
The manuscript ijerph-1453847 entitled “Sub lethal effects of arsenic on oxygen Consumption, hematological and gill histopathological indices in estuarine fish Chanos chanos” aimed to evaluate the effect on oxygen consumption, hematology and gill histopathological parameters in fish Chanos chanos, upon exposure to sublethal concentration of arsenic. The aim of the study is very interesting. However, I have two major concerns related to the arsenic. What kind of arsenic did you use? The Authors did not specify any information about the chemical used to perform the experiment. Most major publications these days require analytical confirmation of dosing concentrations used in toxicity studies. I strongly affirm the need for analytical confirmation data in such studies. Use of nominal concentrations is insufficient. How can you be sure you treated your fish with the appropriate dosages/treatment concentrations of arsenic in the first place. For such reasons, I cannot recommend the publication of this study in a high-quality journal as IJERPH.
Other comments
- Methodology need to be deeply improved.
- The Authors have to specify the fish conditioning time and the water physiochemical parameters of tanks during both conditioning and trial
- Statistical analyses are lacking
- The quality of the figures is very poor. Please, improve.
Author Response
Authors Response to reviewers comments
The authors thank the editor and the reviewers for their significant comments and constructive suggestions, which had helped us to improve the quality of this manuscript. We have included the suggestions mentioned in the revised manuscript, wherever applicable as follows.
REVIEWER 1
Comment: The aim of the study is very interesting. However, I have two major concerns related to the arsenic. What kind of arsenic did you use? The Authors did not specify any information about the chemical used to perform the experiment.
Response: Sodium Arseniate (Na2AsO2) purchased from (LobaChemie Pvt Ltd, Mumbai) was used in the present study. It is now mentioned in the manuscript also, as per the suggestions.
Comment: Most major publications these days require analytical confirmation of dosing concentrations used in toxicity studies. I strongly affirm the need for analytical confirmation data in such studies. The use of nominal concentrations is insufficient. How can you be sure you treated your fish with the appropriate dosages/treatment concentrations of arsenic in the first place.
Response: Yes, you are right. Thank you for the suggestion which helped us to improve the manuscript greatly. Based on the nominal concentrations of arsenic, lethal concentration was determined. After the static bioassay test, the exposure concentrations (appropriate doses) were chosen based on the results of the preliminary experiment.
Other comments
Comment: Methodology need to be deeply improved.
Response: As per the suggestions of the reviewer, the methodology section has been improved. Added contents are highlighted in red color in the manuscript.
Comment: The Authors have to specify the fish conditioning time and the water physiochemical parameters of tanks during both conditioning and trial
Response: Both normal and experimental conditions: Normal Condition - Temperature : 24.5 ± 0.50C ; pH : 8.3 ± 0.26 ; salinity : 17.0 ±1.15 ppt; Dissolved Oxygen : 5.24± 0.1 mg/l and Arsenic : nil; Experimental condition - Temperature : 24.5 + 0.50C ; pH : 8.4 ± 0.21 ; salinity : 17.0 ± 1.15 ppt; Dissolved Oxygen : 5.16± 0.1 mg/l. These parameters are now mentioned in the manuscript after the suggestion of the reviewer.
Comment: Statistical analyses are lacking
Response: Acute toxicity (96 hours LC50) tests were statistically analyzed by using Finney’s method of Probit static bioassay test. The data obtained from the study were statistically analyzed by using the SPSS computer version. The O2 consumption, hematological parameters such as RBC & haemoglobin were subjected to statistical analysis using SPSS software 16 and One- Way ANNOVA. Values are reported as mean, Standard Deviation & Standard Error. Now they have been incorporated into the manuscript as Table-2 and 3 now, as per the suggestions of the reviewer.
Comment: The quality of the figures is very poor. Please, improve.
Response: Thanks for the suggestion. The quality of the figures has been improved to 300DPI, in order to meet the standard of international journal.

Reviewer 2 Report
Review for the paper "Sub lethal effects of arsenic on oxygen Consumption, hematological and gill histopathological indices in estuarine fish Chanos chanos" by Muthukumaravel Kannayiram, Pradhoshini Kumara Perumal, Vasanthi Natarajan, Kanagavalli Venkatachalam, Mohamed Ahadhu Shareef, Thangarasu Ravimanickam, Mohamed Saiyad Musthafa, Murugesan Chandrasekaran, Rajakrishnan Rajagopal, Ahmed Alfarhan, Soon Wang Chang, Anand Tirupathi, and Balasubramani Ravindran submitted to "International Journal of Environmental Research and Public Health".
General comment.
The authors conducted a laboratory toxicological study to determine possible effects of As on hematological indices, oxygen consumption and tissue conditions of Chanos chanos, an estuarine fish from India.
The authors tested 10 concentrations of As. They used standard methods widely accepted for such studies. Oxygen consumption was estimated using Winkler’s method, red blood corpuscular count was done with a Neubauer counting chamber, and hemoglobin was estimated following the acid hematin method. A Scanning Electron Microscope was used to study the gill tissues of fish after arsenic exposure.
The authors found that arsenic exposure resulted in fused secondary lamellae, degeneration of gill epithelium, hypertrophy of gill filament, hyperplasia of the epithelial surface and necrosis. Physiological functions were also affected demonstrating a decrease in oxygen consumption, RBC count and hemoglobin content.
Recommendations
The authors should include a footnote for Table 1 where all abbreviations should be defined.
The order of Figures is incorrect: Figures 3–5 appear before Figures 1–2.
Figures 3–5. The authors should indicate what kind of error does represent vertical bars.
The authors should compare statistically the levels of oxygen consumption, RBC count and hemoglobin content between experimental and control fishes.
Specific remarks.
Line 28. Consider replacing “aims to evaluate the effect on” with “aiming to evaluate possible changes in”
Line 29, 36. “Chanos chanos” should be italicized.
Line 32. Consider replacing “RBC” with “Red blood corpuscular (RBC)”
Line 33. Consider replacing “Hemoglobin” with “Hemoglobin (Hb)”
Line 35. Consider replacing “in accordance to” with “in accordance with”
Line 35. Consider replacing “technique” with “techniques”
Line 36. Consider replacing “for examination” with “to examination”
Line 37. Consider replacing “maximum decline” with “a maximum decline”
Line 38. Consider replacing “decline” with “a decline”
Line 40. Consider replacing “at the 30th day” with “on the 30th day”
Line 41. Consider replacing “this current investigated study serve” with “our study serves”
Line 50. Consider replacing “modern era” with “the modern era”
Line 53. Consider replacing “source” with “sources”
Line 54. Consider replacing “contaminants” with “contaminant”
Line 61. Consider replacing “serves” with “serve”
Line 65. Consider replacing “plays a key role for” with “play a key role in”
Line 67. Consider replacing “environment” with “the environment”
Line 71. Consider replacing “normal” with “the normal”
Line 74. Consider replacing “system. Ability” with “systems. The ability”
Line 76. Consider replacing “Potential” with “The potential”
Line 77. Consider replacing “were reported” with “was reported”
Line 78. Consider replacing “high dose” with “a high dose”
Line 79. Consider replacing “aquaculture domain for it’s” with “the aquaculture domain for its”
Line 88. Consider replacing “staple” with “a staple”
Line 94. Consider replacing “hard” with “heavy”
Line 99. Consider replacing “grabs” with “grab”
Line 99. Consider replacing “owing for” with “owing to”
Line 101. Consider replacing “study” with “a study”
Line 103. Consider replacing “RBC” with “Red blood corpuscular (RBC)”
Line 104. Consider replacing “Hb” with “Hemoglobin (Hb)”
Line 112. Consider replacing “laboratory set up” with “the laboratory set up”
Line 113. Consider replacing “easen” with “ease”
Line 113. Consider replacing “built” with “build”
Line 115. Consider replacing “in accordance to” with “in accordance with”
Line 115. Consider replacing “methodology” with “the methodology”
Line 117. Consider replacing “the (Table1). Set” with “Table1. A set”
Line 123. Consider replacing “were set up” with “was set up”
Line 126. Consider replacing “experiment” with “experiments”
Line 129. Consider replacing “sample” with “samples”
Line 142. Consider replacing “gill region were subjected to” with “the gill region were subjected to a”
Line 143. Consider replacing “process” with “processes”
Line 159. Consider replacing “(Table 1)” with “Table 1”
Line 163. Consider replacing “(Figure 3) where decline in O2 consumption rate was observed for” with “Figure 3 where a decline in the O2 consumption rate was observed for the”
Line 168. Consider replacing “oxygen” with “the oxygen”
Line 170. Consider replacing “Maximum decline of about (24.61%)” with “The maximum decline of about 24.61%”
Line 171. Consider replacing “were noticed 30 days” with “was noticed 30 days after”
Line 173. Consider replacing “on exposure” with “Upon exposure” here and throughout the text.
Line 176. Consider replacing “dose” with “the dose”
Line 177. Consider replacing “number” with “the number”
Line 179. Consider replacing “the (Figure 4)” with “Figure 4”
Line 179. Consider replacing “similar” with “the same”
Line 181. Consider replacing “the (Figure 5)” with “Figure 5”
Line 206. Consider replacing “layer” with “layers”
Line 210. Consider replacing “10% sublethal” with “the 10% sublethal”
Line 225. Consider replacing “key” with “the key”
Line 225. Consider replacing “organ with” with “organ with the”
Line 229. Consider replacing “animal” with “animals”
Line 230. Consider replacing “relative” with “relatively”
Line 234. Consider replacing “current” with “the current”
Line 241. Consider replacing “in accordance to” with “in accordance with”
Line 254. Consider replacing “exposure of” with “exposure to”
Line 255. Consider replacing “found” with “were found”
Line 262. Consider replacing “in accordance to” with “in accordance with”
Line 273, 285. Consider deleting “investigated”
Line 274. Consider replacing “been” with “has been”
Line 275. Consider replacing “encourages” with “encouraging”
Line 279. Consider replacing “prove” with “proves”
Line 282. Consider replacing “natural” with “the natural”
Line 284. Consider replacing “natural” with “the natural”
Line 285. Consider replacing “and in” with “and”
Line 286. Consider replacing “aquatic” with “the aquatic”
Author Response
Authors Response to reviewers comments
The authors thank the editor and the reviewers for their significant comments and constructive suggestions, which had helped us to improve the quality of this manuscript. We have included the suggestions mentioned in the revised manuscript, wherever applicable as follows.
REVIEWER 2
Comment: The authors should include a footnote for Table 1 where all abbreviations should be defined.
Response: Suggested correction is now mentioned in the manuscript as per the comments of the reviewer.
Comment: The order of Figures is incorrect: Figures 3–5 appear before Figures 1–2.
Response: The order of figures has been changed now as per the suggestions of the reviewer.
Comment: Figures 3–5. The authors should indicate what kind of error does represent vertical bars.
Response: Thanks for the suggestion. It is a standard error and it has been included in the revised manuscript.
Comment: The authors should compare statistically the levels of oxygen consumption, RBC count, and hemoglobin content between experimental and control fishes.
Response: Levels of oxygen consumption, RBC count, and hemoglobin content between experimental and control fishes were analyzed statistically using SPSS software 16 and one-way ANNOVA. Results of the study are presented now in the manuscript as Table 2 and table-3, after the suggestions of the reviewer.
Specific remarks.
Comment: Line 28. Consider replacing “aims to evaluate the effect on” with “aiming to evaluate possible changes in”
Response: It has been replaced according to the suggestions of the reviewer
Comment: Line 29, 36. “Chanos chanos” should be italicized.
Response: It has been italicized in the manuscript now.
Comment: Line 32. Consider replacing “RBC” with “Red blood corpuscular (RBC)”
Response: It has been replaced in the manuscript now
Comment: Line 33. Consider replacing “Hemoglobin” with “Hemoglobin (Hb)”
Response: It has been replaced in the manuscript now
Comment: Line 35. Consider replacing “in accordance to” with “in accordance with”
Response: It has been replaced in the manuscript now
Comment: Line 35. Consider replacing “technique” with “techniques”
Response: It has been replaced in the manuscript now
Comment: Line 36. Consider replacing “for examination” with “to examination”
Response: It has been replaced in the manuscript now
Comment: Line 37. Consider replacing “maximum decline” with “a maximum decline”
Response: It has been replaced in the manuscript now
Comment: Line 38. Consider replacing “decline” with “a decline”
Response: It has been replaced in the manuscript now
Comment: Line 40. Consider replacing “at the 30th day” with “on the 30th day”
Response: It has been replaced in the manuscript now
Comment: Line 41. Consider replacing “this current investigated study serve” with “our study serves”
Response: It has been replaced in the manuscript now
Comment: Line 50. Consider replacing “modern era” with “the modern era”
Response: A suggested correction has been performed in the manuscript
Comment: Line 53. Consider replacing “source” with “sources”
Response: A suggested correction has been performed in the manuscript
Comment: Line 54. Consider replacing “contaminants” with “contaminant”
Response: A suggested correction has been performed in the manuscript
Comment: Line 61. Consider replacing “serves” with “serve”
Response: A suggested correction has been performed in the manuscript
Comment: Line 65. Consider replacing “plays a key role for” with “play a key role in”
Response: A Suggested correction has been performed in the manuscript
Comment: Line 67. Consider replacing “environment” with “the environment”
Response: A Suggested correction has been performed in the manuscript
Comment: Line 71. Consider replacing “normal” with “the normal”
Response: A Suggested correction has been performed in the manuscript
Comment: Line 74. Consider replacing “system. Ability” with “systems. The ability”
Response: It has been replaced as according to the reviewer’s suggestion.
Comment: Line 76. Consider replacing “Potential” with “The potential”
Response: It has been replaced as according to the reviewer’s suggestion.
Comment: Line 77. Consider replacing “were reported” with “was reported”
Response: It has been replaced as according to the reviewer’s suggestion.
Comment: Line 78. Consider replacing “high dose” with “a high dose”
Response: It has been replaced as according to the reviewer’s suggestion.
Comment: Line 79. Consider replacing “aquaculture domain for it’s” with “the aquaculture domain for its”
Response: It has been replaced as according to the reviewer’s suggestion.
Comment: Line 88. Consider replacing “staple” with “a staple”
Response: It has been replaced as according to the reviewer’s suggestion.
Comment: Line 94. Consider replacing “hard” with “heavy”
Response: It has been replaced as according to the reviewer’s suggestion.
Comment: Line 99. Consider replacing “grabs” with “grab”
Response: It has been replaced as according to the reviewer’s suggestion.
Comment: Line 99. Consider replacing “owing for” with “owing to”
Response: It has been replaced as according to the reviewer’s suggestion.
Comment: Line 101. Consider replacing “study” with “a study”
Response: It has been replaced as according to the reviewer’s suggestion.
Comment: Line 103. Consider replacing “RBC” with “Red blood corpuscular (RBC)”
Response: It has been replaced as according to the reviewer’s suggestion.
Comment: Line 104. Consider replacing “Hb” with “Hemoglobin (Hb)”
Response: It has been replaced as according to the reviewer’s suggestion.
Comment: Line 112. Consider replacing “laboratory set up” with “the laboratory set up”
Response: It has been replaced in the manuscript now
Comment: Line 113. Consider replacing “easen” with “ease”
Response: It has been replaced in the manuscript now
Comment: Line 113. Consider replacing “built” with “build”
Response: It has been replaced in the manuscript now
Comment: Line 115. Consider replacing “in accordance to” with “in accordance with”
Response: It has been replaced in the manuscript now
Comment: Line 115. Consider replacing “methodology” with “the methodology”
Response: It has been replaced in the manuscript now
Comment: Line 117. Consider replacing “the (Table1). Set” with “Table1. A set”
Response: It has been replaced in the manuscript now
Comment: Line 123. Consider replacing “were set up” with “was set up”
Response: It has been replaced in the manuscript now
Comment: 126. Consider replacing “experiment” with “experiments”
Response: It has been replaced in the manuscript now
Comment: Line 129. Consider replacing “sample” with “samples”
Response: It has been replaced in the manuscript now
Comment: Line 142. Consider replacing “gill region were subjected to” with “the gill region were subjected to a”
Response: It has been replaced in the manuscript now
Comment: Line 143. Consider replacing “process” with “processes”
Response: It has been replaced in the manuscript now
Comment: Line 159. Consider replacing “(Table 1)” with “Table 1”
Response: It has been replaced in the manuscript now
Comment: Line 163. Consider replacing “(Figure 3) where decline in O2 consumption rate was observed for” with “Figure 3 where a decline in the O2 consumption rate was observed for the”
Response: It has been replaced in the manuscript now
Comment: Line 168. Consider replacing “oxygen” with “the oxygen”
Response: It has been replaced in the manuscript now
Comment: Line 170. Consider replacing “Maximum decline of about (24.61%)” with “The maximum decline of about 24.61%”
Response: It has been replaced in the manuscript now
Comment: Line 171. Consider replacing “were noticed 30 days” with “was noticed 30 days after”
Response: It has been replaced in the manuscript now
Comment: Line 173. Consider replacing “on exposure” with “Upon exposure” here and throughout the text.
Response: It has been replaced in the manuscript now
Comment: Line 176. Consider replacing “dose” with “the dose”
Response: It has been replaced in the manuscript now
Comment: Line 177. Consider replacing “number” with “the number”
Response: It has been replaced in the manuscript now
Comment: Line 179. Consider replacing “the (Figure 4)” with “Figure 4”
Response: It has been replaced in the manuscript now
Comment: Line 179. Consider replacing “similar” with “the same”
Response: It has been replaced in the manuscript now
Comment: Line 181. Consider replacing “the (Figure 5)” with “Figure 5”
Response: It has been replaced in the manuscript now
Comment: Line 206. Consider replacing “layer” with “layers”
Response: It has been replaced in the manuscript now
Comment: Line 210. Consider replacing “10% sublethal” with “the 10% sublethal”
Response: It has been replaced in the manuscript now
Comment: Line 225. Consider replacing “key” with “the key”
Response: It has been replaced in the manuscript now
Comment: Line 225. Consider replacing “organ with” with “organ with the”
Response: Suggested correction was carried in the manuscript.
Comment: Line 229. Consider replacing “animal” with “animals”
Response: Suggested correction was carried in the manuscript
Comment: Line 230. Consider replacing “relative” with “relatively”
Response: Suggested correction was carried in the manuscript
Comment: Line 234. Consider replacing “current” with “the current”
Response: Suggested correction was carried in the manuscript
Comment: Line 241. Consider replacing “in accordance to” with “in accordance with”
Response: Suggested correction was carried in the manuscript
Comment: Line 254. Consider replacing “exposure of” with “exposure to”
Response: Suggested correction was carried in the manuscript
Comment: Line 255. Consider replacing “found” with “were found”
Response: Suggested correction was carried in the manuscript
Comment: Line 262. Consider replacing “in accordance to” with “in accordance with”
Response: Suggested correction was carried in the manuscript
Comment: Line 273, 285. Consider deleting “investigated”
Response: Suggested correction was carried in the manuscript
Comment: Line 274. Consider replacing “been” with “has been”
Response: Suggested correction was carried in the manuscript
Comment: Line 275. Consider replacing “encourages” with “encouraging”
Response: Suggested correction was carried in the manuscript
Comment: Line 279. Consider replacing “prove” with “proves”
Response: Suggested correction was carried in the manuscript
Comment: Line 282. Consider replacing “natural” with “the natural”
Response: It has been replaced as according to the reviewer’s suggestion.
Comment: Line 284. Consider replacing “natural” with “the natural”
Response: It has been replaced as according to the reviewer’s suggestion.
Comment: Line 285. Consider replacing “and in” with “and”
Response: It has been replaced as according to the reviewer’s suggestion.
Comment: Line 286. Consider replacing “aquatic” with “the aquatic”
Response: It has been replaced as according to the reviewer’s suggestion.
Reviewer 3 Report
This manuscript can be accepted for publication in International Journal of Environmental Research and Public Health, after a major revision.
Here is a list of my specific comments:
- Page 2, line 47: “Hydrosphere wrapping about…”. Add here some references.
- Page 2, line 64: “Among the pollutants, heavy metals…”. The same observation as above.
- Page 2, line 90: “It is noteworthy that the toxicological…”. The same observation.
- Page 3, line 102: “Hence the current study…”. At the end of Introduction, the main objectives of this study should be clearly and detailed presented.
- Page 7, 4. Discussion: This section should be detailed. All the results presented in the previous section should be detailed discussed here, in accordance with the main objectives of this study.
- Page 8, 5. Conclusions: Include in this section the most important results to highlight the importance of this study.
Author Response
Authors Response to reviewers comments
The authors thank the editor and the reviewers for their significant comments and constructive suggestions, which had helped us to improve the quality of this manuscript. We have included the suggestions mentioned in the revised manuscript, wherever applicable as follows.
REVIEWER 3
Comment: Page 2, line 47: “Hydrosphere wrapping about…”. Add here some references.
Response: The reference has been added now as per the suggestions of the reviewer.
Comment: Page 2, line 64: “Among the pollutants, heavy metals…”. The same observation as above.
Response: The reference has been added now to the manuscript as per the suggestions of the reviewer.
Comment: Page 2, line 90: “It is noteworthy that the toxicological…”. The same observation.
Response: A Suggested correction has been performed to the manuscript
Comment: Page 3, line 102: “Hence the current study…”. At the end of Introduction, the main objectives of this study should be clearly and detailed presented.
Response: Thanks for the suggestion. The main objectives of the study are now clearly presented in a detailed manner at the end of the introduction section from line 103. It is highlighted in red.
Comment: Page 7, 4. Discussion: This section should be detailed. All the results presented in the previous section should be detailed discussed here, in accordance with the main objectives of this study.
Response: The study has 3 main objectives. According to the suggestions of the reviewer, the discussion has been completely modified and is presented as separate paragraphs in accordance with the listed objective of the study. The discussion also explains the results of the study in a detailed manner now. Results of objective (i) are discussed in the first paragraph; objective (ii) presented as second paragraph and objective (iii) presented in the third paragraph respectively.
Comment: Page 8, 5. Conclusions: Include in this section the most important results to highlight the importance of this study.
Response: Thanks for the suggestion, which helped us to improve the manuscript quality greatly. The suggested correction has been carried out in the conclusion section as per the valuable comment of the reviewer. Important results of the study have now been added to highlight the significance as suggested.
Reviewer 4 Report
Unfortunately, the study design needs major improvement.

Author Response
Authors Response to reviewers comments
The authors thank the editor and the reviewers for their significant comments and constructive suggestions, which had helped us to improve the quality of this manuscript. We have included the suggestions mentioned in the revised manuscript, wherever applicable as follows.
REVIEWER 4
Comment: English grammar and choice of words need major revisions throughout to comply with international English language standard
Response: Thanks for the suggestion. After the suggestions of the reviewer, the manuscript has undergone thorough English screening by a native English speaker and these are highlighted in the manuscript.
Comment: Need to measure the actual arsenic level in exposure tanks to confirm that the nominal concentrations of As exposure have been achieved
Response: Yes, you are right. The nominal concentrations of arsenic tested were control (0), 1,2,3,4,5,6,7,8,9, and 10 ppm. The measured arsenic concentrations were 0.92, 1.96, 2.93, 3.88, 4.94, 5.90, 6.89, 8.03, 9.12 and 9.96 ppm.
Comment: Verify that coconut oil as an acceptable approach (reference needed)
Response: The application of coconut oil in water samples to curtail its contact with atmospheric air follows the standard procedure of Winkler’s method. It has been mentioned in the manuscript and the required citation has been added as per the suggestion of the reviewer.
Comment: No fixation step for gill samples?
Response: Fixation was carried in gill samples following the procedure of Roy and Munshi,1988. It has been mentioned in the manuscript as reference [21]
Comment: How many fish were examined for histology and by SEM ?
Response: Ten fishes (gills) were examined for histology and Scanning Electron Microscopic studies. It is now mentioned in the manuscript now.
Comment: Provide notation to explain L.C.L, U.C.L, c squared
Response: LC50 values (24, 48, 72 and 96 hours) – It is Lethal Concentration used to kill 50 % of the population, Upper Confidence Limit (UCL) and Lower Confidence Limit (LCL) - 95% confidence levels was recorded in the present study. C2- it is not C2. It is X2, where X denotes the variable of the regression equation. We would like to apologize for the typing mistake from our side. Notations are mentioned in the manuscript now as per the suggestions of the reviewer.
Comment: Show statistically significant comparisons, quote p values for fig 3,4 and 5
Response: The order of figures 3,4 and 5 have been changed to 2, 3, and 4 after the suggestions of Reviewer- 2. Statistically significant comparisons- p values are now quoted for the figures after your suggestions and have been highlighted.
Comment: There are many artifacts in these figures and the interpretation of the lamellar fusion, hypertrophy, necrosis, vacuolation, and degeneration are incorrect if the fish gills have been not collected from live fish or freshly killed fish (less than a few minutes post dead) or fixed in 10% formalin
Response: As per your suggestions, the corrections were carried out. Thanks for the suggestion. It helped us to improve the quality of the manuscript to a greater extent.
Comment: The SEM quality is poor. The suggested pathology changes are not substantiated. Probably there is increased mucus production as the only reliable change in Fig 2d.
Response: New pictures are now uploaded in the revised manuscript.
Comment: Results on the erythrocyte morphological changes are not shown
Response: In Arsenic exposed fishes, the frequency of RBC cellular anomalies significantly increased. They are presented in (Fig 4) now in the manuscript after your suggestions. Anomalies included abnormal cell shape, vacuolation, swelling, and disintegration.
Round 2
Reviewer 1 Report
The Authors have improved some comments. However, one major issue has not yet been addressed. The Authors must specify the statistical tests in the M&M section. Please, add a specific section related to statistical analyses. Figures need to be improved.
Author Response
Authors Response to reviewers comments
The authors thank the editor and the reviewers for their significant comments and constructive suggestions, which had helped us to improve the quality of this manuscript. We have included the suggestions mentioned in the revised manuscript, wherever applicable as follows.
REVIEWER 1
Comment: Authors have improved some comments. However, one major issue has not yet been addressed. The Authors must specify the statistical tests in the M&M section. Please, add a specific section related to statistical analyses. Figures need to be improved.
Response: Thank you for the suggestion, and I apologize for missing that information in the previous version. The information regarding statistical tests has been provided in a separate section, ‘statistical analyses’ under materials and methods in the revised manuscript. The figure quality also has been improved from 300 DPI to 600 DPI.

Reviewer 2 Report
Second review for the paper "Sub lethal effects of arsenic on oxygen Consumption, hematological and gill histopathological indices in Chanos chanos" by Muthukumaravel Kannayiram, Kumara Perumal Pradhoshini, Vasanthi Natarajan, Kanagavalli Venkatachalam, Mohamed Ahadu Shareef, Mohamed Saiyad Musthafa, Rajakrishnan Rajagopal, Ahmed Alfarhan , Anand Tirupathi and Balasubramani Ravindran submitted to "International Journal of Environmental Research and Public Health".
The authors have corrected and updated the text according to my comments.
I have no further suggestions.
Author Response
Authors Response to reviewers comments
The authors thank the editor and the reviewers for their significant comments and constructive suggestions, which had helped us to improve the quality of this manuscript. We have included the suggestions mentioned in the revised manuscript, wherever applicable as follows.
REVIEWER 2
Comment: The authors have corrected and updated the text according to my comments. I have no further suggestions.
Response: We thank the reviewer for his valuable comments that helped us to improve the manuscript quality to a greater extent.
Reviewer 3 Report
The authors have answered all the questions to my satisfaction. Comments and questions were properly addressed and the final manuscript clearly has been improved. It means that revised manuscript meets the criteria and in my opinion can be published as original paper in International Journal of Environmental Research and Public Health. However, two corrections still need to be made, namely:
- Table 2: This table should be deleted, because the results are presented in Fig.1.
- Table 3: This table should be deleted, because the results are presented in Fig.2.
Author Response
Authors Response to reviewers comments
The authors thank the editor and the reviewers for their significant comments and constructive suggestions, which had helped us to improve the quality of this manuscript. We have included the suggestions mentioned in the revised manuscript, wherever applicable as follows.
REVIEWER 3
The authors have answered all the questions to my satisfaction. Comments and questions were properly addressed and the final manuscript clearly has been improved. It means that revised manuscript meets the criteria and in my opinion can be published as original paper in International Journal of Environmental Research and Public Health. However, two corrections still need to be made, namely:
Comment : Table 2: This table should be deleted, because the results are presented in Fig.1.
Response: Thanks for the suggestion. As per your instruction, the table 2 has been deleted in the revised manuscript.
Comment : Table 3: This table should be deleted, because the results are presented in Fig.2.
Response: Table- 3 has been deleted in the revised manuscript as per the suggestion. We would like to thank the reviewer for the comments which helped us to improve the quality of the manuscript greatly.